# Comparing Different Residential Neighborhood Definitions and the Association Between Density of Restaurants and Home Cooking Among Dutch Adults

**DOI:** 10.3390/nu11081796

**Published:** 2019-08-03

**Authors:** Maria Gabriela M. Pinho, Joreintje D. Mackenbach, Hélène Charreire, Jean-Michel Oppert, Harry Rutter, Joline W. J. Beulens, Johannes Brug, Jeroen Lakerveld

**Affiliations:** 1Department of Epidemiology and Biostatistics, Amsterdam UMC, Vrije Universiteit Amsterdam, Amsterdam Public Health, De Boelelaan 1089a, 1081 HV Amsterdam, The Netherlands; 2Université Paris Est Créteil (UPEC), LabUrba, UPEC, 61 Avenue du Général de Gaulle, 94010 Créteil, France; 3Equipe de Recherche en Epidémiologie Nutritionnelle (EREN), Centre de Recherche en Epidémiologie et Statistiques, Inserm (U1153), Inra (U1125), Cnam, COMUE Sorbonne Paris Cité, Université Paris 13, 74 Rue Marcel Cachin, 93017 Bobigny, France; 4Department of Nutrition, Institute of Cardiometabolism and Nutrition, Sorbonne Université, Pitié-Salpêtrière Hospital, 47-83 Boulevard de l’Hôpital, 75013 Paris, France; 5Department of Social and Policy Sciences, University of Bath, Bath BA2 7AY, UK; 6Julius Center for Health Sciences and Primary Care, University Medical Center Utrecht, Huispost Str. 6.131, PO Box 85500, 3508 GA Utrecht, The Netherlands; 7Amsterdam School of Communication Research (ASCoR), University of Amsterdam, Nieuwe Achtergracht 166, 1018 WV Amsterdam, The Netherlands; 8National Institute for Public Health and the Environment, Antoni van Leeuwenhoeklaan 9, 3721 MA Bilthoven, The Netherlands; 9Faculty of Geosciences, Utrecht University, Vening Meinesz building A, Princetonlaan 8a, 3584 CB Utrecht, The Netherlands

**Keywords:** food environment, exposure definition, density of restaurants, cooking at home, adults

## Abstract

The definition of neighborhoods as areas of exposure to the food environment is a challenge in food environment research. We aimed to test the association of density of restaurants with home cooking using four different definitions of residential neighborhoods. We also tested effect modification by age, length of residency, education, and income. This innovative cross-sectional study was conducted in the Netherlands (*N* = 1245 adults). We calculated geographic information system-based measures of restaurant density using residential administrative neighborhood boundaries, 800 m and 1600 m buffers around the home and respondents’ self-defined boundaries (drawn by the respondents on a map of their residential area). We used adjusted Poisson regression to test associations of restaurant density (tertiles) and the outcome ”weekly consumption of home-cooked meals” (six to seven as compared to five days per week (day/week) or fewer). Most respondents reported eating home-cooked meals for at least 6 day/week (74.2%). Regardless of the neighborhood definition used, no association between food environment and home cooking was observed. No effect modification was found. Although exposure in terms of density of restaurants was different according to the four different neighborhood definitions, we found no evidence that the area under study influences the association between density of restaurants and home cooking among Dutch adults.

## 1. Introduction

The role of the environment for food intake and/or related health outcomes has gained increasing attention in the past decade, but the evidence currently available provides mixed results [1,2,3]. The use of varying definitions of exposure to the food environment has been posed as a potential explanation for these inconsistent findings [4]. Defining exposure to the food environment is recognized as a challenge and includes issues such as the choice of food retailers under study (e.g., restaurants, grocery stores, local food shops), but also the metrics used to quantify these retailers, and the definition of the area of interest. 

The area of interest in food environment research is almost always characterized in terms of a neighborhood. According to Chaix et al. (2009), neighborhood definitions can be classified in two main ways: territorial neighborhoods and ego-centered neighborhoods. The first dimension includes neighborhoods defined according to administrative boundaries or legal boundaries at the area level, such as those defined by local governments (e.g., census tracts in the USA). The second dimension includes varied sizes of Euclidian or street network buffers around an individual’s home. The advantage of using ego-centered neighborhoods is that they are more likely to represent an area that individuals move around in, and thus their potential exposure to the food environment in that area [5]. A special kind of ego-centered neighborhoods is the ”self-defined neighborhood”, for which individuals are asked to draw the boundaries of what they regard as their residential neighborhood on a map [4,5,6,7]. If individuals draw these boundaries based on the area they usually spend time in, this definition may provide a more accurate basis for assessment of exposure to their food environment than the alternatives. In turn, a better-defined exposure to the food environment is expected to better capture potential relations between environment, behavioral, and health outcomes [8,9,10]. Nonetheless, challenges with self-defined neighborhoods include large variations in size of self-defined neighborhoods (potentially including implausible neighborhood size), the inclusion of individual variation in an environmental exposure measure, and the impracticalities of assessing self-defined neighborhood boundaries in surveys with large population samples. 

How neighborhoods are defined may impact research findings on the association of (neighborhood) food environments with dietary behaviors because the definition of neighborhood influences the measured exposure to food retailers [11]. A recent systematic review, on methods used to measure the food environment in relation to obesity, found that the methods used are very diverse and poorly reported. In addition, many associations were dependent on the methods used [12]. Therefore, to explore implications of using different neighborhood definitions, we studied the associations of characteristics of the food environment according to four neighborhood definitions with frequency of home cooking. Home cooking has been associated with better diet quality indicators such as lower energy intake; lower consumption of fat, sugar and ultra-processed foods; higher adherence to dietary guidelines; and higher consumption of fruit and vegetables [13,14,15,16]. Less time spent on home-cooking preparation, in turn, has been associated with more visits to fast-food restaurants and more money spent on eating out [17]. Indeed, a previous study, using data from the European SPOTLIGHT survey, showed that lower neighborhood access to restaurants, but not greater access to grocery stores, was associated with a higher frequency of home cooking. Therefore, exposure to restaurants may be an important environmental determinant of home cooking, and in the current study we hypothesize that lower neighborhood access to restaurants is associated with a higher frequency of home cooking. Previous studies has also indicated that the relationship between environment and health behaviors may be modified by sociodemographic characteristics such as age, income, and education [18,19]. In addition, an earlier study we conducted on self-defined neighborhood boundaries [7] showed that older individuals drew smaller neighborhoods, as well as higher educated, long-term residents and women drew larger neighborhoods, and therefore effect modification by individual characteristics may be anticipated. 

Therefore, in the current study, we explored (1) whether the density of restaurants (including full-service and fast-food restaurants) differs for ego-centered (Euclidean buffers around the respondents’ homes; and respondents’ self-defined neighborhood boundaries) and territorial neighborhood definitions (administrative boundaries); (2) whether a higher density of restaurants was inversely associated with frequency of home cooking and whether this association was stronger for self-defined neighborhoods; and (3) whether these hypothesized inverse associations were stronger for older individuals, lower income and lower educated individuals, and for those who had lived longer in a neighborhood.

## 2. Materials and Methods 

### 2.1. Study Design, Sampling, and Participants

We used data from the cross-sectional European SPOTLIGHT survey, as part of the European SPOTLIGHT project described in detail elsewhere [20]. Data collection took place between February and September 2014 in five urban regions across Europe: Ghent and suburbs (Belgium), Paris and inner suburbs (France), Budapest and suburbs (Hungary), the Randstad (a conurbation including the cities of Amsterdam, Rotterdam, The Hague, and Utrecht in the Netherlands), and Greater London (UK). Detailed information on sampling, study design, and recruitment of participants has been previously published [21]. We randomly sampled 12 neighborhoods in each European urban region, defined by administrative boundaries (60 in total). This sampling was based on a combination of residential density and socioeconomic status (SES) at the neighborhood level. This resulted in four pre-specified neighborhood types: low SES/low residential density, low SES/high residential density, high SES/low residential density, and high SES/high residential density. 

The adults (18 years or older) residing in the selected neighborhoods completed an online questionnaire. The questionnaire was used to assess sociodemographic characteristics, including home address, general health, and lifestyle behaviors including dietary behaviors. In addition, the questionnaire contained a web mapping tool that allowed participants to draw the boundaries of their self-defined neighborhood, i.e., the limits of what they considered to be their neighborhood [7]. We were unable to collect food environment data for all self-defined neighborhood areas, and no harmonized database on the food environment was available across the five countries involved. High quality food environment geographic information systems (GIS) data was available for the Netherlands. Therefore, only data from participants residing in the 12 neighborhoods selected in the Netherlands who drew their self-defined neighborhood boundaries were used for this study, resulting in a final analytical sample of 1245 individuals. The total response rate for the SPOTLIGHT study was 10.8%, and 13.7% among Dutch participants [21]. Detailed information on the selection of participants for this study is provided in Figure 1. The Ethics Committee of the VU University Medical Center in Amsterdam approved the study in the Netherlands. All survey participants provided written informed consent.

### 2.2. Measures 

#### 2.2.1. Exposure to the Food Environment (Independent Variables) 

We used commercially available data on the location of food retailers from Locatus–Locatus, a Dutch commercial company that collects regular information on the location of several types of retail outlets (https://locatus.com/en/) on a national scale. The Locatus staff performed field audits throughout the Netherlands in both rural and urban areas, and recorded location, type, size, and opening times of all retailers. Food retailers located in shopping areas are audited every year; these corresponded to two-thirds of all food retailers. The food retailers located outside shopping areas are audited every 2–3 years and regular checks are performed. The location of food retailers was determined through x and y coordinate points collected in 2014, the same year as the SPOTLIGHT survey data collection. The Locatus dataset was tested against a field audit in selected areas across the Netherlands in 2019. For the validation study, grocery stores (e.g., supermarkets, local food shops, and green grocers) and food outlets (e.g., restaurants, fast food restaurants and cafés) were analyzed separately and showed “good” to “excellent” agreement. For instance, the positive predictive value for location of food outlets ranged from 0.82 for take-away outlets to 0.94 for fast-food restaurants. Values for concordance ranged from 0.76 for full-service restaurants to 0.95 for fast-food restaurants [22].

The food retailers analyzed in this study were classified by the data provider, Locatus, and further categorized into three groups: restaurants, grocery stores, and other food retailers (Table 1). Given our outcome ”frequency of home cooking”, in the restaurant category we included food retailers that most likely sell meals to be eaten away from home such as restaurants, fast food outlets, and kebab shops. In the grocery stores category, we included food retailers that mostly sell ingredients to be prepared at home [23]. In the ”other food retailers” category, we included all food retailers that are not restaurants, i.e., food retailers such as bakery, cheese, and nut stores, but also the food retailers listed under ”grocery stores”. The rationale for this classification was confounding adjustment for the broader food environment. Since we expected that a higher density of grocery stores might provide more opportunities to buy ingredients to cook at home, we tested how adjustment for the presence of grocery stores only influenced our models. Secondly, we tested how the presence of all food retailers available influenced the association between density of restaurants and cooking at home. 

Using ArcGIS^®^ 10.4 (ESRI. Redlands, CA, USA), the count of restaurants was calculated for four different neighborhood definitions: (1) administrative boundaries, as defined by statistics Netherlands (CBS) in 2014, (2) 800 m Euclidean buffers around participants’ homes, (3) 1600 m Euclidean buffers around participants’ homes, and (4) participants’ self-defined boundaries. Figure 2 shows an example of the four different neighborhoods and the food retailers present in the neighborhoods. The buffer sizes were determined based on an assumed distance individuals would, in general, be willing to walk or cycle [24], and commonly used buffer sizes in previous literature [25]. The surface size (in square kilometers) of each neighborhood type was also calculated in ArcGIS. For the four neighborhood definitions used, the count of restaurants was divided by the respective neighborhood area in square kilometers to obtain the density measures. The calculated densities were very skewed and transformations did not result in acceptable distributions. We then decided to use the density of restaurant as categorical variables. We constructed different categorizations, and we opted for tertiles so that the null values for density were mostly placed in the first tertile. However, because there is always a risk that the categorization chosen may lead to a loss of information, and lower the data variability [26], we presented sensitivity analysis using the exposure measures split into quintiles. In order to contribute to a quality report on food environment research, we followed the Geo-FERN (Geographic Information System Food Environment ReportiNg) checklist [27]. The filled check list can be found in the Appendix A.

#### 2.2.2. Cooking at Home (Outcome Variable)

We assessed participants’ frequency of consuming a home-cooked meal with the following questionnaire item: ”How many days a week do you, or does someone in your household, prepare a meal using ingredients as opposed to eating ready or takeaway meals?” The eight response options ranged from less than once a week to seven days a week. In a previous study, using data from the five urban regions we divided frequency of home cooking into three categories [28]. However, among Dutch participants only, the first category (0–3 times per week) contained only 71 participants. Therefore, we divided the outcome variable into two categories: low frequency of home cooking (0–5 days per week, *n* = 257) and high frequency of home cooking (6–7 days per week, *n* = 790).

### 2.3. Confounding Variables 

Sociodemographic covariates used in this study to control for confounding effects were age, sex, educational attainment, income, employment status, and household composition. Following the Dutch Standard Classification of Education (SOI), the nine response options for self-reported educational attainment were categorized into three groups: ”lower” (ranging from no education to general secondary education)’ ”medium” (secondary vocational and higher general secondary education), and ”higher” (university degree or higher). The five response options for net household income were aggregated into three categories: lower (€2000 per month or less), medium (€2000 to €2800 per month), and higher (€2800 or more). Employment status was categorized into ”yes” (individuals either employed or in education) or ”no” (homemakers, those retired or unemployed). Household composition was divided into “1 person”, “2 persons”, or “3 or more persons”. A potential causal association between exposure to the food environment and diet may be distorted in observational studies due to selection effects. Since it is virtually impossible to conduct experimental studies that can measure the effect of the food environment on dietary behaviors, it is very important that observational studies adjust for self-selection bias, which is an important source of bias in food environment research [29,30]. We controlled for potential self-selection bias by adjusting our models for the following three dichotomous neighborhood confounding variables: (1) whether participants spent most of their leisure time in their neighborhood, (2) whether individuals had lived in their neighborhood for 10 years or longer, and (3) whether they had given the presence of restaurants as a reason for choosing to live in the neighborhood. Food retailers often colocate, and different types of retailers may, therefore, simultaneously influence the food behavior of individuals [31]. Adjustment for the broader food environment was done in two steps: Adjustment for the presence of grocery stores only, and then adjustment for the presence of grocery stores and other food retailers such as cheese stores, bakeries, and pastry shops. 

### 2.4. Statistical Analyses 

The percentage of missing values ranged from 0.3% for the “employment status” variable to 20.8% for the variable ”presence of restaurants as a reason for choosing the neighborhood”. Participants with missing values were on average older (59.1 vs. 51.1 years). A higher percentage of females (56.6% vs. 51.1%), lower educated (21.0% vs. 11.8%), and lower income (34.4% vs. 26.1%) individuals was found for participants with missing data as compared to those with complete data. Since the missing values did not appear to be randomly distributed across categories, missing values were assumed to be missing at random instead of being missing completely at random. We used the predictive mean matching method to perform multiple imputation on all variables in the analysis [32]. Given the percentage missing values we imputed 30 datasets as recommended by Rubin [33] and Bodner [34]. Pooled results from the 30 inputted datasets were used in the regression models and non-inputted data were used to perform the descriptive analysis. 

In order to test for clustering at the neighborhood level, we added a random intercept for administrative neighborhood groups. Using 2-log likelihood ratio tests, we found no significant clustering at the neighborhood level, for any of the four neighborhood definitions, and therefore no multilevel analysis was performed. Since our dichotomous outcome variable, i.e., frequency of home cooking 6–7 days per week, was frequent (74.2%), we used Poisson regression with robust variance [35]. To test the association of density of restaurants and frequency of home cooking, we used three different models for each neighborhood definition. Model 1 had the density of restaurants (in tertiles) as an independent variable and was adjusted for sociodemographic and the neighborhood self-selection variables. Model 2 was adjusted similar to Model 1 with the addition of density of grocery stores as a continuous variable and Model 3 was adjusted similar to Model 1 with the addition of density of all other food retailers as a continuous variable. Because an earlier study we conducted on self-defined neighborhood boundaries [7] showed that older individuals drew smaller neighborhoods, and higher educated, long-term residents and women drew larger neighborhoods, therefore, we tested effect modification by age groups (<65 year vs. ≥65 year, i.e., the retirement age in the Netherlands); educational attainment; level of income; and years of residency in the neighborhood by adding multiplicative interaction terms to all the models separately.

We performed some sensitivity analyses that are presented in the supplementary material. First, we ran our main analysis including only self-defined neighborhoods that were within the range of area sizes for administrative neighborhood boundaries (0.25 to 4.14 km^2^, *N* = 720). This was done because the range for self-defined neighborhoods as drawn by individuals varied greatly in size (from 0.02 to 443.2 square kilometers). Therefore, we wanted to explore if results of analyses with self-defined neighborhoods were influenced by or attributable to extreme self-defined neighborhood sizes. A second sensitivity analysis we performed was descriptive statistics for participants excluded from the full sample. We also present in supplementary files the coefficients for the associations between the study covariates and home cooking, as well as coefficients for joint effects of exposures studied and potential effect modifiers and their product term as a third sensitivity analysis. Fourth, we performed sensitivity analyses of the main models using the exposure measures split in quintiles. 

We considered the absence of the value 1 in the 95% confidence intervals to be a statically significant association, but interpreted our results mainly based on the magnitude of the effect sizes. Multiple imputation, descriptive, and regression analyses were conducted in STATA 14^®^ (StataCorp. College Station, TX, USA, 2015).

## 3. Results

Table 2 presents the characteristics of the total sample, as well as by groups defined by frequency of home cooking. The majority of the participants who indicated that they cook six to seven times per week (74.2% of total sample) were female (56.2%), higher educated (64.5%) and had a higher income (55.7%). In unadjusted analyses (Table 2), there was a consistent trend for an association between higher density of restaurants and lower frequency of home cooking, regardless of the neighborhood definition used. 

Table 3 shows the neighborhood characteristics. The surface size of the 12 neighborhoods defined by administrative boundaries ranged from 0.25 to 4.14 square kilometers (km^2^), with a median of 0.62 km^2^. The median density of restaurants in neighborhoods defined by administrative boundaries was 0.78 (interquartile range (IQR) 0–18.7). The median density of restaurants within the 800 meter buffer was 4.97 (IQR 1.49–8.95), 3.23 (IQR 2.24–8.33) for the 1600 m buffer, and 4.07 (IQR 0–19.6) for the self-defined neighborhoods. 

Table 4 shows spearman correlation coefficients for the density of food retailers in the four neighborhood definitions used. Almost perfect correlations (ρ = 0.9) were found for densities of restaurants, grocery stores, and other food retailers within a 1600 m buffer. Almost perfect correlations were often found as well between density of other food retailers and grocery stores. The lowest correlation values were found for administrative neighborhood as compared to the other measures used. 

Table 5 shows the results for the association of density of restaurants by four different definitions of neighborhood with a weekly frequency of home cooking. Regardless of the neighborhood definition used, the density of restaurants was not associated with the frequency of home cooking and the effect sizes were negligible. Sensitivity analyses (Appendix A), performed only with individuals who drew their self-defined neighborhood boundaries in a size ranging from 0.25 to 4.14 km^2^, showed similar results to those presented in Table 4. We further tested for effect modification by age, education, income, and years of residency in a neighborhood but no effect modification was found.

Appendix A presents descriptive characteristics of the SPOTLIGHT participants according to the exclusion criteria presented in Figure 1. Dutch participants were on average older, and Dutch participants who drew self-defined neighborhood boundaries showed the highest frequency of cooking at home: 74.1% those participants cooked six to seven days per week as compared to 65.0% of the full sample. Appendix A presents effect sizes for associations between frequency of home cooking and all the study covariates. Different neighborhood definitions were not important for the association between the covariates and home cooking as the effect sizes were all similar across the four different neighborhood definitions tested. Appendix A presents the associations of joint effects of exposures and the potential effect modifiers and their product term with frequency of cooking at home. The model used to perform these analyses was model three (adjusted to all covariates and additionally adjusted for the density of all other food retailers). Appendix A shows sensitivity analysis performed as the main models, but with the exposure measures split in quintiles. Results were comparable to the main analysis using tertiles, where regardless of the neighborhood definition, the density of restaurants was not associated with the frequency of home cooking. 

## 4. Discussion

We studied whether the use of four different definitions of residential neighborhood was related to the association between density of restaurants and frequency of home cooking among Dutch adults. Exposure to density of restaurants differed across the neighborhood definitions, but this did not translate into different associations with home cooking. Regardless of the residential neighborhood definition, density of restaurants was not associated with frequency of home cooking and no relevant effect sizes were observed. Finally, no effect modification was observed for age, education, income or years of residence in a neighborhood.

Descriptive neighborhood statistics showed a large variation of area sizes and total number of food retailers across neighborhoods. Regarding densities of food retailers, we observed that for ego-centered neighborhoods, i.e., the 800 m and 1600 m buffers around residential address, as well as self-defined neighborhoods, the median density of restaurants was more similar than the median density of restaurants within administratively-defined neighborhoods (territorial neighborhood definition). This was also demonstrated in the correlation analysis between the exposure measures used, where the lowest correlation values were found for administrative neighborhoods as compared to the other measures. This is in line with the work of Burgoine et. al. (2013) who compared how different measures of exposure to the food environment correlated. They used both density and proximity measures and found that these measures were strongly correlated in several instances, however, when comparing those metrics across distinct neighborhood classifications (i.e., area level neighborhood definitions and individual buffers), the correlations between these two measures were not very strong [11]. The correlation analysis also showed almost perfect correlations for all exposure measures within a 1600 m buffer. In this case, the existence of a threshold effect of exposure may also be considered. It is possible that the presence of food retailers within a 1600 m buffer is more evenly distributed across the foodscape than when using smaller buffer sizes. Thus, it may not be possible to detect variation in the sample. Very strong correlations were also found for the density of other food retailers and grocery stores. This finding is plausible given the fact that part of the food retailers present in the other food retailers category are also present in the grocery stores category. 

Hobbs et al. (2017) tested the association between the availability of food retailers and BMI using different neighborhood definitions (an administrative neighborhood definition vs. different buffers sizes) in the UK. They found differences in the number of food retailers across different neighborhood definitions, but only minor differences were observed in the associations between food retailers and BMI [36]. This finding is puzzling since we expected that a change in the exposure, i.e., differences in the density of restaurants across neighborhood definitions, would translate into a difference in the associations. It could be that the observed variation in the total number of food retailers across neighborhood definitions was not sufficient to detect any association, or that exposure to restaurants does not influence individuals’ cooking behaviors. 

The lack of significant association between the density of restaurants and the frequency of home cooking in the present study is in contrast to our previous observation that, using the complete cross-European SPOTLIGHT population, higher spatial access to restaurants was associated with lower frequency of home cooking [28]. One potential explanation for this lack of association may be a lower variability in the dichotomous outcome measure used for these analyses, as compared to the three categories used in the analysis with the complete sample. This is also evident in the sensitivity analysis showing descriptive statistics for participants excluded from the full sample. This analysis showed that a higher percentage of Dutch participants reported cooking six to seven days per week as compared to the full sample, possibly leading to a lower variation in the outcome variable. It could also be that the Dutch population has different characteristics from the wider study population. To test that, we replicated the analysis used in the cross-European study using only data of participants from the Netherlands. This showed that spatial access to restaurants was indeed not associated with frequency of home cooking among Dutch participants, which led us to conclude that the presence of restaurants may not be important for the frequency of home cooking within the Dutch population. The definition of home cooking may also be different among the Dutch and the European sample. The somewhat crude questionnaire question used to evaluate the frequency of home cooking could perhaps not have captured nuances in the definition of home cooking by the participants. However, there is no validated measure to assess the frequency of home cooking in surveys. Questionnaire items inquiring how frequently individuals or family members prepared home-cooked meals are commonly used in the literature [14,15,16,37]. As a consequence, this lack of association may explain why we were not able to find differences in the associations for the different neighborhood definitions. The definition of neighborhood may be important for associations between other health behaviors and outcomes, but this would need to be further investigated. 

This study is innovative for being the first to explore four different definitions of residential neighborhoods, uniquely including a participant-defined neighborhood definition, and also for testing effect modification by individual level factors. The uncertain geographical problem is a well-known limitation on environmental research. This concept demonstrates how the measured influence of area-based characteristics such as the density of food retailers on an individual’s health behaviors is dependent on the way neighborhoods are defined [38]. Many methods used to define neighborhoods do not accurately represent the area individuals move around in. By analyzing participants’ self-defined neighborhoods, that may better represent their activity space, and also by comparing both ego-centered and administratively-defined neighborhoods, this study contributes to the literature base. Although we did not address temporal dynamics in the food environment, we did account for space variation by using a self-defined neighborhood [39]. This work builds on previous research on individual and environmental level determinants of home cooking, which have led to important methodological decisions such as to control our models for self-selection bias and other individual level determinants of home cooking, and the use of relative measures of exposure to the food environment [28,29,30,40,41].

Limitations include the fact that we could not adjust for exposure to other food environments, such as work or leisure settings [42]. Information on where people actually go for grocery shopping and consumption of meals, as well as within store audits and information on the consumer environment (e.g., price, promotions, and range of choices) would also have been desirable [43,44,45]. The use of a specific determinant (density of restaurants) and specific outcome (frequency of cooking at home) may limit the generalizability of these findings, especially in the light of lack of associations in this Dutch subsample of the SPOTLIGHT project. Finally, the low response rate for the SPOTLIGHT survey (around 10%) and the selection of Dutch participants for this study might have produced a selective sample and, therefore, the possibility of selection bias should be considered. The SPOTLIGHT study used a stratified random sampling approach and informed participants that the study was about their neighborhood (i.e., not emphasizing the interest in lifestyle and overweight). As such, the total sample, as well as the Dutch subsample, is well balanced in terms of sociodemographic characteristics such as sex, education, income, and BMI, but other biases may remain. For example, it might be the case that people who are highly influenced by unhealthy food environments were less likely to participate than the general population. 

## 5. Conclusions

In conclusion, the definition of area under study does not seem to matter for the association between density of restaurants and cooking at home within the Dutch adult population. However, due to the lack of consistency in food environment research and the important role the definition of exposure plays in that matter, whether the area under study is important for determining exposure to other aspects of the food environment and/or other behavior and health outcomes should be further investigated.

## Figures and Tables

**Figure 1 nutrients-11-01796-f001:**
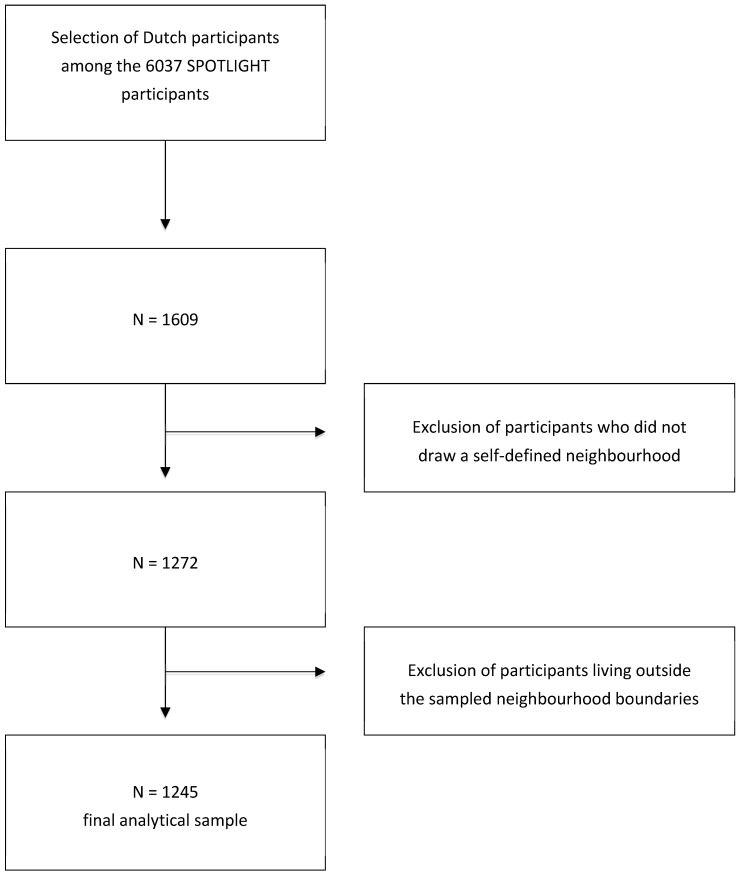
Sample selection.

**Figure 2 nutrients-11-01796-f002:**
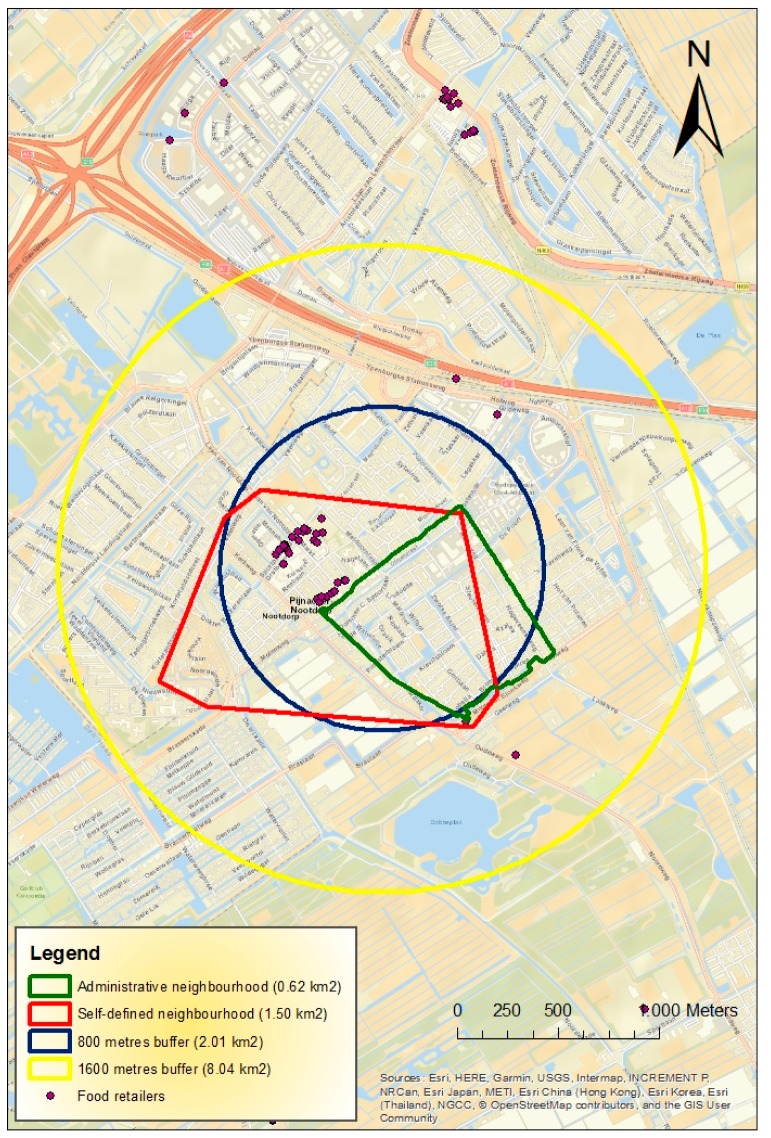
Example of the four neighborhood definitions and the food retailers present.

**Table 1 nutrients-11-01796-t001:** Categories and definitions of food retailers analyzed. Definitions are as established by Locatus, the dataset provider.

Analytical Category	Composed of	Food Retailers’ Main Provision of Foods:
Restaurants	Fast food chains and locally owned fast food restaurants such as kebab stores	Mostly deep-fried products that are ready for consumption in few minutes after ordering. Usually there is no table service available.
	Food delivery and Take away	Meals that are not consumed in the store, but are collected or delivered
	Full-service restaurant	Provision of meals a-la-carte, table service is present. Drinks are only provided in combination with food
	Café-restaurant	Provision of both drinks and simple meals
	Restaurant in hotels	Overnight in combination with an a-la-carte restaurant
	Pancake stores	Restaurant specialized in pancakes and/or crepes
Grocery stores	Supermarket	Store selling a wide range of food and non-food products which are used on a daily basis. Store size should be at least 150 m^2^
	Small grocery store	Same as supermarkets but store size is less than 150 m^2^
	Greengrocers	Potatoes, vegetables and fruit
	Butchery	Meat and meat products
	Poultry shop	Poultry
	Fish stores	Fish, crustaceans, and molluscus
	Natural/organic food store	Organic foods and products supplemented with nutritional supplements, homeopathic products and herbs
	Oriental food store	Shop that mainly sells oriental food
Other food retailers	Grocery stores *	See above
	Bakery	Bread and pastries. Table service is possible, but this is not be the main store activity
	Pastry shop	Pastries, pies and cakes
	Chocolate shop	Chocolate, normally of higher quality
	“Delicatessen”	Special and high-end foods and often also many ready-made products
	Cheese store	Cheese
	Nuts store	Nuts
	Confectionery store	Candies and chocolates

* All the food retailers in the ”grocery stores” category were also included in the ”other food retailers” category.

**Table 2 nutrients-11-01796-t002:** Descriptive characteristics of the Dutch SPOTLIGHT participants by total sample and according to the frequency of cooking at home.

Characteristics	Total Sample		Frequency of Cooking at Home (Days Per Week)		
			0–5	6–7	
	*N*		25.8%	74.2%	*p*-Value
Age—mean (SD)	1239	54.0 (15.8)	52.7 (15.8)	53.7 (15.6)	0.352 ^a^
Sex (%)	1236				0.001 ^b^
Female		52.8	44.8	56.2	
Educational attainment (%)	1157				<0.001 ^b^
Lower		14.4	16.8	13.0	
Medium		26.1	34.0	22.8	
Higher		59.5	49.1	64.2	
Income	1094				<0.001 ^b^
Lower		28.2	36.5	24.9	
Medium		21.5	25.6	19.4	
Higher		50.4	38.0	55.7	
Household composition (%)	1150				<0.001 ^b^
1 person		25.8	43.1	17.8	
2 persons		41.7	35.4	44.9	
3 or more persons		32.4	21.5	37.3	
Employed or in education (%)	1242				0.055 ^b^
Yes		60.5	66.6	60.2	
Spend most of spare time in the neighborhood	1224				0.326 ^b^
Yes		76.1	73.5	76.4	
Presence of restaurant is a reason for living in the neighborhood	986				<0.001 ^b^
Yes		13.7	21.1	10.7	
Years of residency in the neighborhood	1232				0.888 ^b^
Less than 10 years		35.8	36.0	36.4	
10 or more years		64.2	64.0	63.6	
Densities of restaurants according to:					
800-m buffer around residence	1245				<0.001 ^b^
T1 (lowest access)		34.5	28.2	36.2	
T2		34.9	31.3	36.0	
T3 (highest access)		30.6	40.5	27.9	
1600-m buffer around residence	1245				0.001 ^b^
T1 (lowest access)		38.3	29.6	41.1	
T2		28.4	29.3	27.9	
T3 (highest access)		33.3	41.2	31.0	
Administrative neighborhood boundaries	1245				0.001 ^b^
T1 (lowest access)		34.9	33.0	35.8	
T2		36.8	30.3	38.4	
T3 (highest access)		28.3	36.7	25.8	
Self-defined neighborhood boundaries	1245				0.004 ^b^
T1 (lowest access)		35.7	29.3	38.2	
T2		30.9	29.9	30.6	
T3 (highest access)		33.3	40.8	31.2	

^a^ ANOVA; ^b^ Chi-square; IQR = interquartile range. T1, T2, and T3 are tertiles of densities, where individuals in T1 have the lowest density of restaurants and individuals in T3 the highest density.

**Table 3 nutrients-11-01796-t003:** Differences in counts and densities of food retailers across neighborhood definitions.

	Neighborhood Area (km^2^)		Restaurants		Grocery Stores		Other Food Retailers	
			Count	Median Density (IQR)	Count	Median Density (IQR)	Count	Median Density (IQR)
	Min–Max	Median (IQR)	Min–Max		Min–Max		Min–Max	
**800 m buffers**	2.00	-	0–133	4.97 (1.49–8.95)	0–53	3.48 (1.49–5.47)	0–65	5.47 (2.49–9.45)
**1600 m buffers**	8.00	-	0–642	3.23 (2.24–8.33)	0–150	2.11 (1.12–3.73)	0–233	3.11 (1.49–5.47)
**Administrative neighborhoods**	0.25–4.14	0.62 (0.34–1.28)	0–41	0.78 (0.00–18.70)	0–14	2.93 (0.00–11.56)	0–19	2.93 (0.78–14.13)
**Self-defined neighborhoods**	0.02–443.26	0.55 (0.21–1.33)	0–2325	4.07 (0.00–19.62)	0–632	3.93 (0.00–12.69)	0–897	5.83 (0.00–18.97)

IQR: Interquartile range.

**Table 4 nutrients-11-01796-t004:** Spearman correlation coefficients for the measures of density of food retailers in the four neighborhood definitions used.

		Restaurants				Grocery Stores				Other Food Retailers			
		(1) *	(2)	(3)	(4)	(5)	(6)	(7)	(8)	(9)	(10)	(11)	(12)
Restaurants	800 m buffer (1)	1.0											
	1600 m buffer (2)	0.8	1.0										
	Self-defined neighborhood (3)	0.7	0.7	1.0									
	Administrative neighborhood (4)	0.8	0.8	0.7	1.0								
Grocery stores	800 m buffer (5)	0.8	0.8	0.7	0.7	1.0							
	1600 m buffer (6)	0.7	0.9	0.7	0.7	0.9	1.0						
	Self-defined neighborhood (7)	0.6	0.6	0.8	0.5	0.7	0.7	1.0					
	Administrative neighborhood (8)	0.5	0.5	0.5	0.6	0.7	0.7	0.6	1.0				
Other food retailers	800 m buffer (9)	0.8	0.7	0.7	0.7	1.0	0.8	0.7	0.7	1.0			
	1600 m buffer (10)	0.7	0.9	0.7	0.7	0.9	1.0	0.6	0.6	0.8	1.0		
	Self-defined neighborhood (11)	0.6	0.6	0.8	0.6	0.7	0.7	1.0	0.5	0.7	0.7	1.0	
	Administrative neighborhood (12)	0.7	0.7	0.6	0.8	0.9	0.9	0.7	0.9	0.9	0.9	0.6	1.0

* The numbers in parenthesis represent the measure and the respective food retailer as demonstrated in the second column of the table.

**Table 5 nutrients-11-01796-t005:** Incidence rate ratio (IRR) and 95% confidence intervals (95% CI) as derived from Poisson regression analyses indicating the associations between density of restaurants, according to four different definitions of neighborhoods, with weekly frequency of cooking at home among adults in the Netherlands. The SPOTLIGHT Project (*n* = 1245).

		Frequency of Home Cooking (6–7 days per week)		
Density of Restaurants		Model 1	Model 2	Model 3
		IRR (95% CI)	IRR (95% CI)	IRR (95% CI)
800 m buffers	T1 (lowest)	1	1	1
	T2	1.01 (0.94–1.09)	1.04 (0.96–1.12)	1.04 (0.96–1.12)
	T3 (highest)	0.95 (0.86–1.05)	1.05 (0.93–1.19)	1.04 (0.91–1.18)
1600 m buffers	T1 (lowest)	1	1	1
	T2	0.94 (0.87–1.02)	0.95 (0.88–1.03)	0.95 (0.87–1.03)
	T3 (highest)	0.94 (0.86–1.03)	0.99 (0.89–1.10)	0.98 (0.88–1.09)
Administrative neighborhoods	T1 (lowest)	1	1	1
	T2	1.00 (0.93–1.08)	1.02 (0.94–1.10)	1.02 (0.94–1.10)
	T3 (highest)	0.97 (0.87–1.07)	1.02 (0.92–1.13)	1.03 (0.91–1.17)
Self-defined neighborhoods	T1 (lowest)	1	1	1
	T2	0.99 (0.91–1.06)	0.99 (0.91–1.06)	0.98 (0.91–1.06)
	T3 (highest)	0.97 (0.89–1.06)	0.98 (0.88–1.08)	0.96 (0.86–1.07)

T1, T2, and T3 are tertiles of densities, where individuals in T1 have the lowest density of restaurants and individuals in T3 the highest density; Model 1 adjusted for age, sex, education, income, household composition, employment status, spare time spent in the neighborhood, years of residency in the neighborhood, and presence of restaurants was a reason for choosing the neighborhood. Model 2 additionally adjusted for the density of grocery stores; and Model 3 additionally adjusted for the density of all other food retailers. IRR = Incidence rate ratio.

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
