# Peer review of "Comparing Different Residential Neighborhood Definitions and the Association Between Density of Restaurants and Home Cooking Among Dutch Adults"

_nutrients, 2019, doi:10.3390/nu11081796_

Round 1

Reviewer 1 Report

This study examines four different types of neighbourhood definitions and then investigates associations between density of restaurants and associations with cooking at home. Surprisingly, in contrast to previous results using the full sample this study shows no association; this may suggest results of the full sample may vary through space. This study provides a contribution to evidence that the authors should be commended for. Well done. Despite this, improvements to the manuscript can be made throughout. I hope that my points for further clarification are considered as points for constructive feedback. I have three major points and the rest are relatively minor suggestions.

Important points to note

-          The prevalence of food eaten out of home has increased substantially over the years. Therefore, it is plausible that no association was found due to authors investigating the association between the prevalence of food establishments providing food to eat out of the home but, measuring an outcome of food consumption within the home. If authors investigated the association between food outlets providing food to eat outside the home and food consumed outside the home would this not be the hypothesised pathway? Could the authors provide a conceptual map (or similar) to show in more detail why this association is hypothesised to occur or at least clarify within the introduction particularly with the focus on restaurants.

-          Please consider including a sensitivity analyses for study participants lost to missing data or for who data could not be collected. Were they different by sample characteristics? From an original sample of 6,037 to 1,245 is a around 21% of the sample left. More details on who was excluded and if this may bias results is an essential amendment even if included in supplementary online material.

-          While data were categorised into tertiles please report a sensitivity analyses for quartiles and quintiles if possible. The trouble with tertiles has been reported previously here by Lamb et al. 2015 https://ijbnpa.biomedcentral.com/track/pdf/10.1186/s12966-015-0181-9?site=ijbnpa.biomedcentral.com

Other minor concerns:

Abstract

-          Line 29 please be consistent with terminology. Either use relevant areas of exposure OR neighbourhoods.

-          Line 30 is it worth stating your population are adults?

-          Line 34 were administrative boundaries based on home address – please explicitly specify

-          Would suggest referring to ‘four types of neighbourhood definitions’ throughout rather than various definitions which is somewhat vague. You have administrative, 800m 1600m and self-drawn as far as I can see?

-          Line 40 please make reference to the scale/magnitude of the effect rather than significance.

-          I appreciate the abstract wording is tight however, I would suggest adding a short sentence on your original contribution explicitly. You have many novel aspects which should be highlighted.

Introduction

-          Please consider updating your references to include the most recent systematic review on the food environment by Wilkins et al. 2019 A systematic review employing the GeoFERN framework to examine methods, reporting quality and associations between the retail food environment and obesity. Health and Place.

-          This more recent systematic review will also allow you to update the work of Chaix in 2009 in the second paragraph as it thoroughly examined study methods, quality and reporting.

-          Please provide more detail on why modification was expected by age, income and education. I have provided several studies which may help develop the context around this:

-          https://www.ncbi.nlm.nih.gov/pubmed/29983271

-          https://www.mdpi.com/1660-4601/14/11/1290

-          If possible to include a Figure may help set up the context of the study for instance by presenting an example of the four different definitions of neighbourhood you use.

Methods

-          Suggest a new paragraph in methods line 110.

-          Please modify your methods to include all the details requested in the GEO-Fern reporting framework see here:

-          https://www.sciencedirect.com/science/article/pii/S1353829216302799

-          Table 1 is somewhat confusing for instance, grocery stores are the main heading for column 2 yet grocery stores are included in other food retailers not grocery stores. Please expand on this perhaps provide examples and the codes used to extract the data from the database for international readers. Following the GEO-Fern reporting framework should allow this.

-          Again I would try to go with ‘four definitions’ of neighbourhood not ‘various’ as this is vague.

-          Evidence is required around the size of radial buffer used to support 800m and 1600m as walking and cycling distance.

-          Did lack of variation of low frequency cooking at home lead to little variability in an important exposure? More details on how this may have impacted on your results is required (Line 170)

Results and discussion

-          Please remove Lines 239 to 241 as these appear to have been left in here by accident. In addition, a proof read of the whole manuscript would benefit readability.

-          Would subheadings help guide the reader in the results?

-          Please remove lines 290 – 292 as these also appear to have been left there by mistake.

-          Please add an explicit sentence to the discussion highlighting your novel contribution to evidence.

-          In the reference list 15 (Wolfson) appears to be one space to the left too many relative to the other references.

Reviewer 2 Report

The article is important and new developments of residential neighbourhood definition is important and worthwhile to be published. However, the aim of the article is very broad looking into three aims mentioned under introduction (more information below), but different topics are not presented carefully. The article would benefit from dividing in two articles. One more methodical oriented article, which is connected to the different forms of neighbourhood definition and one about the association between restaurant density and home cooking.

Aim 1) Whether Density of restaurants differ considering different forms in defining residential neighbourhood. This is a very important aim. It is nicely deduced in the introduction. However, no results are presented and the discussion about this topic is scarce. Please show similarity or differences between the presented four different forms of neighbourhood definitions.

Aim 2) Association between restaurant density and home cooking meals. This topic is very shortly deduced in the introduction, results are presented but the discussion should go more in depth.

Aim 3) About potential interaction: it is wrong formulated At the moment it is written: Whether these “inverse association”… However, there is no association. Therefore it is tricky to already suppose a result before having it. Please rewrite this aim or delete it.

To my understanding the interaction in a not existing association is unimportant. I like to suggest an aim about different personal predictors (like sex age, SES, or education and or income) that might explain variation in ego-centred neighbourhoods, This aim fits to the first aim but not to the second one.

 Furthermore, I have some problems in your choice of “Restaurants” you put together. Fast-food and Kebab stores and restaurants with a dining room. In my opinion these are different types of restaurants. In a fast food restaurant, you are going because you do not like to cook. It is really a substitution of home cooking. A dining restaurant is another type where you celebrate some special events, or you have enough money to pay for it. It is not really a substitution of home cooking but more a supplement of home cooking. Furthermore, the quality of food is different. Fast food restaurants serve fatty foods with a lot of energy and less amount of vegetables. However, in a dining restaurant you can also find health food. Furthermore, when thinking about your confounders, you should have in mind that low income will lead to use more fast food and less dining restaurants. Therefore, your adjustment is not adequate as you put different restaurant types that are differently correlated with your potential confounders in the same category. I am not surprised that you do not find the suggested association; a more in-depth analysis is necessary. This should be considered in the analyses or at least discussed.

Introduction:

The introduction is nicely written however it over pronounce methodical topics (Aim 1) and Aim 2 is only fairly introduced.

Method description.

The method description is wired. It shows that the writer knows a lot about the used method. But it is not reader friendly. There is too much unimportant information. But sometimes the important information is still is missing. A method part is like a cake receipt. You need to describe what you do. In a cake receipt, you do not like to read a medical explanation why eggs are unhealthy. You need the simple information: Put two eggs into a bowl. The method part is similar. It confuses the reader when they need to read what was not done and why. Please start the paragraphs with the information what was done and then discuss why?

Response rate is missing.

I do not get the explanation why the three neighbourhood confounding variables might control for self-selection bias (line188ff). I believe the mentioned confounders are important as they explain/adjust for participant’s interest in living in this area. However, you do not adjust for self-selection.

Result

Please delete the first paragraph of the result section (Line 239-241). It is senseless.

Otherwise, the results are nicely presented. However, to answer the first aim some information is missing. For the first aim, I would like to have a comparison between all four different forms of neighbourhood definition. How are they correlated? Do the calculated tertils of restaurant densities given the different neighbourhood categorisations defines similar groups?

Discussion:

Please delete the first paragraph of the result section (Line 290-292). It is senseless.

The discussion needs more content related discussion. Why it is important to talk about the association between density of restaurants and home cooking. Please try to think about the sociology in living in such a neighbourhood and its effect on home cooking. Why this correlation should be there? Why you do not find it? You discuss a lot about the statistical measurements, but you miss to discuss your considered exposure (density of restaurants). Why it was chosen? What does it explain? What are the predictors for using a dining restaurant or using a fast food restaurant?
